# Early Childhood Teachers' Confidence to Teach Religious Education and the Influences Which Impact Their Teaching of Religious Education in Catholic Primary Schools

Sharon Law-Davis

School of Education, The University of Notre Dame Australia, Fremantle 6160, Australia;
sharon.law-davis1@nd.edu.au

**Abstract:** This article reports findings from a study that examines the factors which influence early childhood and care (ECEC) early career teachers' confidence in teaching Religious Education (RE) and how it impacts their teaching of RE in Catholic primary schools in Western Australia. Teachers' confidence is a teacher's belief in their ability to accomplish a goal and refers to strength of belief. Research in this area has shown that confidence is correlated with the sense of teaching efficacy and responsibility for student learning. Whilst there has been some research into teacher confidence, there has been insufficient research into early childhood teachers' confidence to teach RE in Catholic schools in an Australian context. Literature supports the notion that if early childhood teachers have a strong teacher confidence for a subject, they are more motivated to teach the subject and this has a positive correlation with positive student outcomes. The epistemological approach underpinning the research is constructivist in nature; therefore, it is based on understanding the constructed realities of what humans know of the world and themselves as produced by communications and systems of meaning. Three case studies follow teachers from their early career to second-year experience in Catholic schools. This study investigates support structures and aspects that contribute to teacher confidence in the teaching of RE. The main themes identified by the data that contributed to confidence or lack of confidence included training, family and religious backgrounds, teaching and learning, mentoring and support in the first year of teaching. The implications of the results for pre-service teacher training and support for graduate teachers are discussed and some suggestions are offered about the courses provided by universities and what schools and education systems can do to support early career teachers to teach RE effectively in Catholic schools.

**Keywords:** confidence; early childhood and care; religious education; teacher education; early career teachers; confidence to teach religious education

## 1. Introduction

Teacher confidence is the ability of a teacher to accomplish goals and refers to their skills, knowledge and belief in their capacity to achieve these goals. The purpose of this study was to ascertain the factors which contributed to early career early childhood education and care (ECEC) teachers' confidence to teach Religious Education (RE) in Catholic schools in Western Australia (WA). Although there has been some literature on the confidence of teachers to teach in other learning areas, there is limited research and literature on the experience of early career ECEC teachers regarding their confidence to teach religious education (Noseda 2020). Teachers' confidence is understood as a teacher's belief in their ability to accomplish a goal and refers to strength of belief (Bandura 1993). Research in this area has shown that confidence is correlated with the sense of teaching efficacy and responsibility for student learning (Hobson and Malderez 2006).

A previous study by Law-Davis and Topliss (2023) explored teacher confidence in the primary school context but did not differentiate between early childhood and primary

teachers. This research is connected to the previous study and early childhood participants were recruited from the previous study, which intended to drill down into the teacher's experience over time through semi-structured interviews. Semi-structured interviews allowed teachers to express their experiences of teaching RE in the classroom at three different stages of their early careers and the aspects that might have contributed to their confidence. Within the interviews, there were perceived to be consistent factors which contributed to the early career teachers' confidence or lack of it. The epistemological approach underpinning the research is constructivist in nature; therefore, it is based on understanding the constructed realities of what humans know of the world and themselves as produced by communications and systems of meaning (Braun and Clarke 2013). The ECEC teachers who participated in this study attended the only Catholic university in WA in their aspiration to teach RE in early childhood classrooms in Catholic schools. Literature supports the notion that if early childhood teachers have a strong teacher confidence for a subject, they are more motivated to teach the subject and this has a positive correlation with positive student outcomes (Pendergast et al. 2011). Noseda's 2020 report on the experiences of pre-service teachers in Australia gives insight into courses that would increase pre-service teachers' confidence to teach RE due to an increased understanding of the mandated curriculum, application of the curriculum, process of teaching and learning and reflective practice. Law-Davis and Topliss (2023) study into teacher confidence to teach RE in Catholic primary schools found that teachers were confident when they had a strong religious background and intensive training in a well-designed curriculum. Law-Davis et al.'s (2019) study into teacher confidence to teach Scripture in Catholic schools found that training in specific RE pedagogy gave early career teachers confidence to teach in the RE learning area. This current study looks at the specific developmental stage of early childhood in Catholic primary schools, children from 5 years to 9 years of age, and the early career teachers' lived experiences, such as family background, religious background, and schooling to discover whether this was influential in their confidence to teach RE in the classroom. Furthermore, this study aims to discover the aspects of teacher training that influenced their confidence to teach RE and how Catholic schools support them in their early years of teaching RE. The participants were early career teachers in their second year of teaching in Catholic schools when the final interview was completed. Three ECEC teachers agreed to be interviewed early in their first year of teaching and their statements were used to support the research. The teachers were interviewed further in their career, in their second year of teaching, and comparative findings will be reported in this paper. During the final interview, participants were questioned about their confidence and plans for their future in teaching RE in Catholic schools.

The purpose of interviewing the early career teachers at different stages of their career is to understand their confidence in teaching RE and whether confidence persisted over time. The study aimed to uncover the aspects that influenced the confidence of the early career teachers of RE Samples of the three interviews are in the Appendix A. In theory, teachers' confidence should increase with proper and timely mentorship from experienced and trained mentors and guidance from leaders and other more experienced teachers (Topliss 2020). Lived experience and appropriate professional learning (PL) can increase teacher confidence and as the literature supports, teachers who are confident in a learning area tend to teach it and build skills in teaching effectively in the learning area (Klassen and Chiu 2010). This research aimed to discover whether this is accurate in the case of three early childhood teachers in their early career. The researcher explored scholarship in the following areas: curriculum, pedagogy and professional learning in relation to mentorship and self-confidence to teach RE.

### 1.1. Curriculum

Studies have shown that teachers who are trained in using a well-designed curriculum and successfully implement it with a range of strategies, such as exploring other perspectives and enriching social problem solving, will improve outcomes for their students

(Department of Education and Training 2015; Kim 2016). Subsequently, teachers who do not have training in an RE curriculum or receive expert mentoring and professional learning in the learning area, do not build their skills to teach RE (Huth et al. 2021). Expert mentoring is required for teachers to reflect on their practice and effectiveness in teaching RE (Topliss 2020). Together, systemic documents, such as a mandated curriculum and policies (or executive directives), assist graduates with their decision making regarding planning, teaching and evaluating learning. Several studies support the notion that curriculum and reflective practices, together with an experienced mentor, ensure consistency with early childhood practice and have been shown to lead to improvements in the teaching and learning of RE, which has positive outcomes for the students (Carroll-Lind et al. 2016; Jones et al. 2016). Grajczonek (2017) emphasizes the practice of providing a well-designed curriculum that is underpinned by the teachers' understanding of theory and appropriate pedagogy to improve student outcomes in RE. Catholic Education Western Australia (CEWA) provides a mandated RE curriculum for the early childhood, primary and secondary years. The mandated curriculum or Units of Work include teacher background materials, sample teaching and learning plans, resources, assessment, and judging standards. The CEWA also have mentoring programs for early career teachers and early career days for teachers in their first year of teaching RE in a Catholic school.

*1.2. Pedagogy*

Teachers make decisions for the effective teaching of RE, decisions that are intentional and thoughtful about the curriculum and strategies that enhance student learning and positively influence learning outcomes (Boyd and Cutcher 2015; Kilderry 2015). Pedagogical decisions influence the way the curriculum is implemented but also influence the environment in which the students are engaged in. Both physical and social spaces need to reflect the needs of the learner, such as opportunities for problem solving, peer learning, collaboration and visual and digital learning opportunities (Department of Education and Training 2015). Teacher training in early childhood pedagogy influences the strategies that early career teachers use to teach RE. Play is essential for children to create knowledge (Grajczonek 2017). Through interactions with others through play, children can construct meaning. Studies on play support the theory that play develops children's creative-thinking skills and teachers can support them in their thinking and enhance their learning (Grajczonek 2017; Leggett and Newman 2017). Teacher training in specific early childhood pedagogy and RE is necessary for teachers to feel confident with the materials and knowledgeable about the RE content. Godly Play is a strategy for teaching Scripture in RE lessons and is especially effective for early childhood settings. Godly Play is invitational to players and includes the language of God and God's people (Berryman 2006). Through concrete materials, the storyteller tells sacred stories, parables and liturgical actions. Silence and wondering are key aspects of Godly Play where those gathered can wonder about the stories being told and create using their imaginations.

*1.3. Professional Learning*

When appointed to a teaching position in Catholic school, teachers complete accreditation to teach RE in a Catholic school to 'gain an understanding of the Catholic faith, tradition and practice in order to carry out their particular responsibility' (Catholic Education Western Australia 2021). The CEWA has established rigorous pathways for the preparation and ongoing professional learning and formation of staff. In their teacher training at university, students require three tertiary courses in theology and two courses in RE methodology to teach RE in Catholic schools in WA to start the accreditation process, which is completed in their first year of teaching RE in a Catholic school. The pathways are achieved through the accreditation framework that offers professional learning programs. Accreditation to teach RE in Catholic schools is renewed every five years after completing 30 hours of knowledge and faith formation components. The accreditation process estab-

lished and required by the CEWA supports early career teachers' confidence to teach RE in Catholic schools.

### 1.4. Self Confidence

Early childhood educators' confidence to teach RE, which includes their beliefs and skills in teaching, is important as it translates to their effectiveness in knowledge and pedagogy to teach RE to improve students' skills, knowledge and understanding (Newton et al. 2012). Teacher self-confidence and efficacy can be viewed as the teacher's capacity to effectively educate children (Bandura 1993); it is a belief that they have the skills required to develop success in children's learning. Teachers who have high confidence in teaching have higher expectations for student achievement, which tends to translate to better student outcomes. Teacher self-confidence can vary across content areas; teachers' comfort with RE and their family and religious backgrounds with teacher training can influence their pedagogical self-confidence and efficacy. Newton et al state that teaching RE is content-specific and requires self-confidence to teach this learning area effectively (2012).

Klassen and Chiu (2010) found that teachers' self-efficacy increases with experience from early career to mid-career. Research by Erden and Sönmez (2011) has indicated that early career teachers hold more positive attitudes towards using developmentally appropriate strategies to teach. This may be because students have been exposed to current pedagogical strategies in ECEC in their university courses. It could also be that early career teachers engaged in mastery experiences, which Bandura describes as the 'most authentic evidence of whether one can muster whatever it takes to succeed' (1997).

## 2. Results

Early childhood teachers were asked to reflect on factors that impacted their confidence to teach RE during the first two years of their teaching following training. The results are presented in three time periods or stages. Stage 1 provides an analysis of the reflection of the early career teacher's experiences in the second term of their first year of teaching. Stage 2 analyses the early career teachers' experiences toward the end of their second year of teaching RE in a Catholic school, and Stage 3 analyses beliefs at the end of the two years of teaching and future directions of the participants. Within the interviews there were consistent themes that emerged which contributed to the early career teachers' confidence. These four main themes identified by the data included curriculum and pedagogy, training, family and religious backgrounds, and mentoring and support.

All three interviews will be combined for each stage and for each theme within each stage.

### 2.1. Curriculum and Pedagogy

Stage One: Curriculum and Pedagogy

The early childhood teachers in stage one of this project identified the units of work provided by the CEWA as being a significant and positive influence in their confidence in the early stages of teaching in a Catholic school. Being provided with a well-designed curriculum gave the teachers a starting point and a sequence of what to teach (content); suggested teaching and learning plans provided by the CEWA were valued and essential to establish the early career teachers' confidence. Questions in the first interview such as "describe your experience in your training that helped build your confidence", (Appendix A) gave participants opportunity to describe specific situations or activities in the training that built their confidence to teach RE.

In the first interview, Gemma explained that she was able to access the RE units of work before she went on an internship, and the RE course at university set an assessment which gave her the confidence to explore the curriculum.

> When it came to assessment time (at university) we had to do the big FPD (forward planning document) and we looked at the units of work and we could really hone in but when I went on prac I could use that FPD that I generated . . . it

made me feel really comfortable with that and we had lots of opportunities during our tutorials to explore the whole curriculum and I think that was really beneficial.

Sophia expressed similar experiences using the RE units of work from the CEWA in the university methodology course, which gave her confidence to teach RE in the first year, stating:

I have been able to teach the units so that's good to have that experience. I found them really helpful, obviously with all the studies we had in uni exploring the units of work . . . it gives us the work to focus on.

Mary also felt that the RE units of work were influential in her confidence to teach RE in the first year as an early career teacher:

RE was one of the subjects that I felt I was well prepared for, especially the RE units of work were so straight forward in what to program and what the expectations of the learning points and activities was expected (sic). So I found that RE was my fun lesson that I always knew it would be a fun, crafty, full of good conversation sort of lesson, especially for Year One. So I looked forward to it, especially at the start.

In the second stage of this project, the early childhood teachers reported a different perspective in using the RE units of work.

Stage 2: Curriculum and Pedagogy

In the second stage, the three early career teachers reported using the CEWA units of work in a more reflective way. They commented on the importance of the RE units of work from the CEWA as continuing to support them; however, as they have gained experience, their reliance on them was not so important and not as strictly prescribed. The three teachers interviewed stated they had less reliance on the suggested learning and teaching plan and more confidence to plan lessons that were more suitable to the students in their class. They were starting to plan lessons to cater for diverse learners in their classroom. One teacher commented that their class really engaged well with discussions and enjoyed sharing their wonder questions as a class. She also commented that her class was very social, and she planned many interactive and collaborative activities to suit their learning style.

These three teachers stated that they were more discerning in the use of the resources and they were more aware of the students that they were teaching. They would often adapt the resources to suit the students and they perceived that they were more effective in the teaching of RE in their classroom.

Gemma described how she used the resource,

I was able to use them (the RE units of work) but to take it one step further and we can do this with this, we can link this with this because I know it so well . . . . I do not need to rely on what somebody else has written for me and you can use those teachable moments . . . I was able to bring those elements into my teaching as well.

This was also experienced by Sophie, who stated,

I feel quite confident now it is my second-year teaching Year One and I have taught across the curriculum content areas for the second year. I can do things the same or change the things that worked and extend things because I am more knowledgeable on them so quite good. The units of work are quite supportive in helping me in planning.

Stage 3: Curriculum and Pedagogy

While the curriculum initially had a major influence in teaching, at the end of the two years, there was not a reliance on the curriculum. Early career teachers were stating

that they were able to achieve 'flow', where the RE curriculum was integrated into other learning areas. Sometimes this took the teacher by surprise, as Gemma reported,

> The curriculum really flows, it works really well with their social/emotional development as well and fits in with all the other areas so sometimes I'm a bit surprised when I have gone to reinforce what I have already planned but as I am reinforcing it and teaching the lesson, I think 'oh my gosh, it links perfectly to what we spoke about'.

At this stage, pedagogy was becoming more important to them; they were becoming more confident to explore the arts and specific RE pedagogy, such as Godly Play, to teach RE, as their statements confirm:

Sophie describes how early childhood pedagogy impacted RE lessons in this statement:

> I find I integrate RE with the arts, with dramatic play, bring in the literature a lot . . . we do Godly Play, we do lots of visual arts to go with it and we link our school values to a lot of the learning in RE.

Similarly, Sophie commented on integrating faith, life, and culture into RE lessons, expanding on the curriculum, "teaching RE is something I believe in and to be able to teach kids to speak up on what you believe in and speak up about your faith and relate it to our own lives and do things in action rather than just listen."

The early career teachers were commenting on decisions they were making as more experienced teachers to increase student engagement in RE. Early childhood pedagogy was implemented using the mandated curriculum. The teachers in the research commented on how they made the learning experiences developmentally appropriate and related to the students' lives to extend their understanding of the content. When asked about how effective they are in teaching RE, Sophie commented:

> I feel like it is because the kids are quite engaged in the stories and resources we write and experiences that are hands on and related to real life and being able to extend on things the students are learning and are able to take to different levels.

Stage One: Training

The three teachers interviewed in stage one identified both the RE methodology and Theology courses within their training as having value for them as early career teachers. The university courses that taught them how to use the CEWA units of work were identified as having the most impact on their confidence to teach RE. They used the units of work sample teaching and learning plans, judging statements and teacher background materials. Each of the early career teachers mentioned these in their statements:

Sophie expressed her experience of the university training:

> I found the methods courses were most valuable because they helped me with understanding the Theology courses which were helpful for the background of our faith.

Georgia supported Sophie's statements, adding that the early childhood courses taught at university gave her the confidence to teach RE using early years pedagogy and skills to provide a learning environment for all children:

> The early years units were very useful in teaching RE. Play and Pedagogy and Learning Environments were really useful because they all aligned. I already had that background and understanding when I started my RE methods courses.

Stage Two: Training

The early career teachers in stage two of this research still consider the training they received at university as the most influential factor in their confidence to teach RE in Catholic schools. However, they have also received some professional learning (PL) within the Catholic school to support their teaching of RE in the classroom. The teachers have said that this has also supported their ability to teach effective lessons in RE.

Sophie summed up her experience:

> We learnt so much (at university) and we learnt how to use them (RE units of work) and find the Godly Play and we had PL on that this year at school so that was great. We pick up quite a lot as we have PL every year. Some of the PL is about knowledge (content) and some are RE, more Godly Play and we had one on the units of work as well, so we had people come in to refresh on them.

Similar findings were noted in the next stage of this project. The early career teachers in their second year of teaching found that their initial training and the PL provided by the school supported their development and confidence to teach RE in the classroom.

Stage Three: Training

In the final stage of the project, the three teachers still used the resources and knowledge obtained at university to teach RE. Together with the PL provided by the Catholic school, they felt confident that their effectiveness as RE teachers would continue to grow.

Mary commented that the RE method course provided her with the methodology, strategies, and resources that she continued to use in the teaching of RE in her classroom:

> How to teach it especially some of those resources you gave us, I still do Godly Play in my classroom I still use Celebrating with Children resource and some of the websites I use all the time.

Sophie had a similar response when asked what aspect was most influential on her confidence to teach RE; she stated:

> Probably the founding I had at university because without that you can have as much knowledge in the world but without being shown the resources and the process of teaching religious education, I don't think I would be as effective.

Two out of the three early career teachers expressed that they were actively applying for further study at the university that they trained to further their qualifications in RE. As summarized by Sophie:

> I am heading back to uni, I have just been accepted to do my Masters in religious education. I love religious education so that's why I wanted to come back and do my Masters. I have considered teaching in university.

### 2.2. Family and Religious Background

Stage One: Family and religious background

Prior experience and family background were significant influences in the confidence of the early career teachers in teaching RE in early childhood classrooms. The quality of their own Catholic education was influential in their ability to teach RE effectively in their own classrooms. Having a religious background, such as being baptized a Catholic in a strong Catholic family, was a positive influence in their confidence to teach RE. In the second interview, questions such as "Last time we spoke you talked about your background can you attribute anything in your background that might have contributed to your experience teaching RE" (Appendix B) this question built from the statements in the first interview where students talked about the influence of their family and the advantage of being raised in a Catholic family that gave them confidence to teach RE. This question gave participants opportunity to be specific and give examples of how family and religious background influenced their confidence to teach RE.

A snapshot of the teachers' statements follows:

> I have received a Catholic education from 3-18. As I have learnt most of the content I will teach through my own education, that obviously influenced my readiness and confidence. (Mary)

> I've been in the Catholic system my whole life and my family is Catholic, so this is how I was raised I don't really shy away from the tricky questions. (Gemma)

> I went to a Catholic primary school and high school, so I feel quite confident with the main stories and the topics and background. Also, I experienced it myself so that definitely helped. My parents and especially my grandparents were regular at attending Mass. (Gemma)

Stage Two: Family and religious background

The early career teachers at the second stage of this project expressed that the experiences that they have had in their own lives within Catholic schools continued to be influential as they teach in early childhood classrooms.

Gemma has explained how her experiences continue to give her the confidence to teach RE:

> I have to say, I have been lucky, that I have had positive experiences being taught myself in Catholic schools and then learning how to teach religious education . . . instead it has always been a very positive experience in all aspects of my own life when it comes to be taught at uni, or at school and then for me, teaching it. It is why I am happy teaching it, I feel quite confident so I feel confident teaching it.

Stage Three: Family and religious background

In the final stage of this project, the early career teachers provided examples of how their family and religious backgrounds influenced how they viewed their responsibility as Catholic teachers. They credited their religious background and religious experience as being influential on how they lived their lives and how their experiences influenced the integration of faith and life; as Sophie stated, "I guess I am more interested in volunteering because of my RE background that gave me opportunities for community service like the soup kitchen and homeless care."

*2.3. Mentoring and Support*

Stage One: Mentoring and support

When interviewed initially, the three early career teachers felt supported at the Catholic school where they were teaching by the leadership team and other teachers. They stated that they received support from the CEWA in their first year of teaching in the form of mentoring support. All three teachers attended an induction course as part of the accreditation requirements at the CEWA and they were involved in the early career professional development days in their first year of teaching. They found the RE professional learning (PL) provided by the CEWA useful and that it supported their confidence in teaching RE. Gemma stated that the pre-primary teachers who attended the PL days created a network on social media where they would chat for support in their early years of teaching:

> Last week I enjoyed it so much because a lot of the teachers were Pre-Primary teachers and like me, in a one-stream school and now we have created a little facebook page where we can network with each other.

Stage Two: Mentoring and support

At the second stage of this project, the early career teachers felt that they had sufficient support in the first year of teaching RE in the Catholic school. Both the school and the CEWA had provided timely and effective mentoring for them in their first year of teaching. They all stated that they felt confident that they could receive assistance within the staff of their school and that nurturing respectful relationships with their colleagues was vital to supporting each other in the school.

Gemma stated that she was offering support to other teachers, for example, by assisting more experienced teachers to find the CEWA units of work, assisting teachers in writing staff prayers, and directing teachers to suitable RE websites. This highlights the confidence she has gained from her experiences.

Stage Three: Mentoring and support

The three early career teachers have credited mentoring and support from the staff and leadership in the school as being a significant influence in their confidence to teach RE in a Catholic school. Opportunities to network with other teachers and combining school PL was also beneficial to developing skills to teach RE. Gemma summarizes this experience:

> To be fair I really think it comes down to a really supportive working environment where you have the opportunities to share your ideas and how you are tracking in class because I think that is what comes with being in a really small school, a one stream school, we are really close and feel able to be confident to share that.

When asked about the networking group they formed at the CEWA PL days, the early career teachers agreed that this was not as necessary now that they had formed professional relationships in their own schools and were feeling supported and confident to teach RE in their classrooms.

### 2.4. Future Directions

In the final interview, the early childhood teachers were also asked what their future directions were. Interview questions such as "Can you tell me a little about where you are in your career?" and "What plans do you have for your future career?" (Appendix C) delve into the participants' early career experiences and how their future careers can develop, giving insight into their confidence to teach RE.

All three teachers had a positive outlook for their future teaching RE in a Catholic school. Two out of three teachers were in the process of applying and being accepted into a postgraduate degree at the university where they trained.

One stated, "I am heading back to uni. I have just been accepted to do my master's in religious education. I love religious education so that's why I wanted to come back and do my Masters. I have considered teaching RE at University".

The early career teachers in this research were leading others in RE in their respective schools. Gemma described the experience of taking a lead in her school:

> I am like the 'go to' person now for planning staff prayers, I have written prayers for other staff members to use. I have shared resources with the staff, things like website you can go to find prayers for different occasions.

## 3. Discussion

A significant finding in this research shows that the early career teachers participating in this project were well-prepared to teach RE; therefore, they were confident and effective in their teaching of the content. This research found that the university courses were effective in preparing early childhood teachers to teach RE in Catholic schools. All three teachers interviewed stated that the university courses, the methodology courses in particular, were of most value in influencing their confidence to teach RE.

The mandated curriculum provided by the CEWA was essential in supporting teachers' knowledge of the content and in providing a teaching plan to follow and suggested resources. An important finding was that as the teachers gained experience and confidence in teaching RE, they were able to cater for the needs of their students to adapt the curriculum for effective teaching. They used developmentally appropriate pedagogy to teach RE in early childhood classrooms through play-based learning and integration of the curriculum.

Another significant finding was that the three teachers identified their own family, schooling and religious backgrounds as being influential in building their confidence to teach RE. It was the experience of being brought up in a Catholic family, being taught in a Catholic school and being trained in a Catholic university that gave them the confidence to teach RE in a Catholic school. They were immersed in a Catholic 'embrace' where they knew the stories, the essential content and experienced the rituals which allowed them to feel confident to share their experience and feel confident in teaching RE. This is an important finding that supports Hackett and Lavery (2010) recommendation that students who do not have the prior knowledge and family background may need essential faith

formation through PL and mentoring to teach RE confidently. This could be a focus of future research where quantitative and qualitative methods could be employed.

Finally, a significant finding was that when students were in environments that they perceived as supportive, they were more confident in teaching RE. The three teachers agreed that the mentoring received at different stages of their experience was influential in supporting their confidence to teach RE. Some examples identified were mentoring during their training and practicums; once employed in a Catholic school, the early career program provided by the CEWA and the mentoring provided by other teachers and leadership teams were instrumental in supporting their confidence to teach RE.

Early career teachers in this research were invited to participate from the cohort of graduate teachers from the RE methodology course. Those who volunteered were obviously confident in what they were doing and enthusiastic to share their experience with the researcher. This would not be the experience for many teachers of RE in the early years. This study is significant to ascertain the factors that produce confident and effective early childhood RE teachers. The findings also acknowledged the importance of various aspects that build confidence in RE teachers. Importantly, the results can be used to decide how schools and systems can support early career RE teachers to continue their work in Catholic schools.

## 4. Materials and Methods

The early career teachers in this study had completed a four-year undergraduate programme to achieve a Bachelor of Education Early Childhood and Care qualification. The research used a storyline study to investigate the early childhood teachers to ascertain their confidence to teach RE during the first two years in their early career. This involved interviewing three participants to collect data about early childhood teachers' perception of their confidence and aspects that influence their confidence to teach RE in three different stages in their early career at a Catholic school.

Three early career teachers were interviewed at different stages of their teaching careers. In stage 1, the initial interview was conducted in the second term of their first year of teaching. In stage 2, the interview took place toward the end of their second year and looked at the first two years of teaching. In stage 3, the interview looked at the end of the two years of teaching and future directions of the participants. The researcher invited participants who were teaching RE in an early childhood classroom in Catholic primary schools to participate in semi-structured interviews. This qualitative methodology of semi-structured interviews 'gives voice to participants and it probes issues that lie beneath the surface of presenting behaviours and actions' (Cohen et al. 2018) and was deemed to be most suitable. The ethnographic design for this is described as a case study, as the investigator will be using the three participants' case studies. This is also referred to as a storyline study, as the investigator seeks to identify the longitudinal changes in support structures as the participants engage in teaching RE in Catholic primary schools in their early careers. Storyline studies gather data over an extended period of time; it can be short-term or long term (Cohen et al. 2018). This is appropriate methodology as the students are engaged in the same experience of teaching over time. There are some disadvantages to using this methodology, such as low participation rate and interviewing being a time-consuming process.

To begin the study, the cohort of early childhood teachers from the course Principles of Primary RE 2 was invited to participate in the study via their university email addresses. A participant information sheet and consent forms were included in the initial email. If the participants agree to be interviewed, they returned the signed consent form or indicated via email that they would like to be part of the study. Participants provided their email addresses to the investigator. The participants who agreed to be interviewed were sent the interview questions prior to the interview via email. A suitable time was sought to meet via Zoom or iPhone to conduct the interview. The interview was recorded on an iPhone and the recording removed when it was transcribed.

Qualitative data which were collected in the form of open-ended questions were analysed for recurrent patterns and themes, using the six-step process described by Braun and Clarke (2013). This process consisted of familiarisation with the data, coding, searching for themes, reviewing themes, defining and naming themes, and writing up. An inductive coding process was used for analysing the data. Creswell and Creswell (2018), Braun and Clarke (2013) recommend a thematic analysis. This was carried out with the interview transcripts and codes applied. These codes were triangulated by a research colleague and common themes emerged. These themes were refined to four main themes.

## 5. Conclusions

The four main influences identified by the early childhood teachers which developed their confidence to teach religious education in Catholic primary schools included curriculum and pedagogy, training, family and religious backgrounds, and mentoring and support. It was suggested by the pre-service teachers in this research that the CEWA had provided a well-designed curriculum and resources to support their teaching of RE. Early career days were organized for those in their first year of teaching in a Catholic school and this was seen as being very supportive and useful to the early career teachers. The CEWA also provided on-going professional learning and early career teachers found the networking within these courses to be especially valuable to them in the teaching of RE.

The conclusions that can be drawn from this research are that early career teachers need specific training in methodology, pedagogy, use of RE curriculum and theology to be confident in teaching RE in Catholic schools. On-going, systematic mentoring of early career teachers is essential for the effective teaching of RE. Mentoring should come from an assigned and trained mentor but can also come from leadership roles and the building of respectful and reciprocal professional relationships within the staff of the catholic school. The training that the early career teachers received at university was a significant aspect in their confidence to teach RE. The theology courses together with methodology courses provided the early career teachers with the skills, knowledge and understanding of the RE learning area. The early career teachers' family and religious backgrounds should not be underestimated. Experience in a Catholic family and schooling provided the early career teachers with confidence to teach RE. Teachers who do not have a Catholic background in their family and in their education may require further mentoring and professional learning to develop the skills necessary to support the teaching of RE in Catholic schools. Research into early career teacher backgrounds and their unique needs could be the focus to further ascertain the targeted mentoring needed to support teachers of RE. Investigating the efficacy of teachers who show confidence in teaching RE and those who do not would be interesting for further research. A larger participation group is recommended with a range of backgrounds, experiences and profiles which would give the research more insight into a wider range of early career teacher's experiences teaching RE in Catholic schools.

**Funding:** This research received no funding.

**Institutional Review Board Statement:** The study was approved by the Ethics Committee, University of Notre Dame Australia Approval Number 09159F date of approval 30/08/2022.

**Informed Consent Statement:** Informed consent was obtained from all subjects involved in the study.

**Data Availability Statement:** Data are available from the author by written request.

**Conflicts of Interest:** The author declares no conflict of interest.

## Appendix A

Interview 1
Title of Research
Early Childhood Teachers' confidence to teach religious education and the influences which impact their teaching of religious education in Catholic primary schools
Interview Questions

Could you tell me a little bit about your school, how long you have taught for and the year level you are teaching?

Do you teach RE in your classroom?

Describe your experience teaching RE.

Could you attribute anything in your background that might have contributed to your experience of teaching RE?

Discuss the contributions of your education? Family background? Religious background? Training to teach RE? University degree? Mentoring? Practicums? Teachers and leaders?

Do you feel confident to teach RE in your classroom?

What priority does the school place on teaching RE?

What priority do you place on teaching RE in your class?

Is your teaching effective in RE?

How do you know?

In your early career in this school, have you had any support? Please discuss.

Do you have any support for teaching RE in your class? Discuss.

**Appendix B**

Interview 2

Title of Research

Early Childhood Teachers' confidence to teach religious education and the influences which impact their teaching of religious education in Catholic primary schools

Interview Questions

Could you tell me a little bit about your school, how long you have taught for and the year level you are teaching?

Do you teach RE in your classroom?

Describe your experience teaching RE.

Last time we spoke you talked about your background can you attribute anything in your background that might have contributed to your experience teaching RE.

What are the foundation skills that you brought with you that supported you through your early teaching career to teach RE in the classroom?

What was the most valuable skill that you brought with you to teaching RE?

Do you feel confident to teach RE in your classroom?

What priority does the school place on teaching RE?

What priority do you place on teaching RE in your class?

Is your teaching effective in RE?

How do you know?

In your early career in this school, have you had any support? Please discuss.

Do you have any support for teaching RE in your class? Discuss.

**Appendix C**

Interview 3

Title of Research

Early Childhood Teachers' confidence to teach religious education and the influences which impact their teaching of religious education in Catholic primary schools

Interview Questions

Can you tell me a little about where you are in your career?

Do you teach RE in your classroom?

How confident do you feel about teaching RE?

As an early career teacher teaching RE in your classroom, do you feel your own skills, knowledge and understanding of RE has improved? In what way and why?

Do you receive any professional learning in faith formation in your role?

What has impacted you the most in developing confidence and efficacy in teaching RE in your early career?

What plans do you have for your future career?

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
