# Peer review of "Early Childhood Teachers’ Confidence to Teach Religious Education and the Influences Which Impact Their Teaching of Religious Education in Catholic Primary Schools"

_religions, doi:10.3390/rel14020198_

Round 1

Reviewer 1 Report

The piece is a very timely piece of work, but it suffers from a number of issues, which I have outlined in the attached file.

Author Response

Thank you for your comprehensive review of this article.  I have taken your advice and made changes as required.  Please see attached document.

Reviewer 2 Report

The three case studies are valuable and interesting. However, the conclusions that can be drawn from them suggest higher potential than fits the research design. A more modest way of concluding and adding more suggestions for quantitative research to be able to draw those conclusions in the future would be very helpful.

Author Response

Thank you for your feedback.  I have attended to your recommendations and will be more discerning when I complete my next article and view my research.

Thank you for your time.

Reviewer 3 Report

See attached.

Author Response

Hello,

Thank you so much for taking the time to review this paper.  I appreciate your constructive feedback and comments to make this paper more effective.  Your feedback on research design and methodology will help me in my future research projects.  I am an early researcher and you have given me many pointers on how I can improve.  Please find attached changes and comments to address your feedback.

Warm regards.

Round 2

Reviewer 1 Report

I think your study has 'legs' and many opportunitues to expand the research and work on a greater logitudinal scope. If you're in Australia, you might link up with Richard Rymarz who does work in this area.  It is going to be important research into the future.

Reviewer 3 Report

Thank you for noting my concerns and taking them on board. I think the conceptual map and a timeline (stating actual dates) could have been included as a graphic n section 4, requiring little explanation.